# Is it feasible to detect FLOSS version release events from textual messages? A case study on Stack Overflow

**Artur Sokolovsky** ***, Thomas Gross, Jaume Bacardit**

School of Computing, Newcastle University, Newcastle upon Tyne, United Kingdom

* b7066789@ncl.ac.uk

**Data Availability Statement:** All the dataset and source code files are publically available from Zenodo database (http://doi.org/10.5281/zenodo.3608827).

**Funding:** Initials of the authors who received the grant are T.G. and J.B. Grant Title is CRITiCaL -

## Abstract

Topic Detection and Tracking (TDT) is a very active research question within the area of text mining, generally applied to news feeds and Twitter datasets, where topics and events are detected. The notion of "event" is broad, but typically it applies to occurrences that can be detected from a single post or a message. Little attention has been drawn to what we call "micro-events", which, due to their nature, cannot be detected from a single piece of textual information. The study investigates the feasibility of micro-event detection on textual data using a sample of messages from the Stack Overflow Q&A platform and Free/Libre Open Source Software (FLOSS) version releases from Libraries.io dataset. We build pipelines for detection of micro-events using three different estimators whose parameters are optimized using a grid search approach. We consider two feature spaces: LDA topic modeling with sentiment analysis, and hSBM topics with sentiment analysis. The feature spaces are optimized using the recursive feature elimination with cross validation (RFECV) strategy. In our experiments we investigate whether there is a characteristic change in the topics distribution or sentiment features before or after micro-events take place and we thoroughly evaluate the capacity of each variant of our analysis pipeline to detect micro-events. Additionally, we perform a detailed statistical analysis of the models, including influential cases, variance inflation factors, validation of the linearity assumption, pseudo $R^2$ measures and no-information rate. Finally, in order to study limits of micro-event detection, we design a method for generating micro-event synthetic datasets with similar properties to the real-world data, and use them to identify the micro-event detectability threshold for each of the evaluated classifiers.

## Introduction

Topic detection and tracking has been an active research question for at least two decades [1]. It focuses on identification and detection of events in text data, like news feeds and Twitter. News articles or tweets in these datasets can be classified as related or not related to the event. The considered events and topics typically include natural disasters, traffic jams, local concerts, celebrity-related events, etc. [2]. The common feature of these events is that they are followed

Combatting cRiminals In The CLoud Grant number: EP/M020576/1 Issued by Engineering and Physical Sciences Research Council (EPSRC) (https://epsrc.ukri.org/) The funders had no role in study design, data collection and analysis, decision to publish, or preparation of the manuscript.

**Competing interests:** The authors have declared that no competing interests exist.

by an observable response from the community, from which the event can be detected. Current study focuses on software engineering (SE)-related forum data analysis. Textual data in SE are present, among others, in the form of Question & Answer (Q&A) platform communications—Reddit and Stack Overflow (SO) to name two examples. These platforms have a large impact on the field because they are commonly queried for code snippets and solutions during the software development process.

Text mining has been used to better understand the influence of the Q&A platforms on SE as well as to improve the coding practices [3]. The study proposes a text mining approach for automatic event detection in forum-like data.

## Research gap

We extend the notion of event by defining micro-events as events, not causing a pronounced response from a linked community and requiring multiple units of text to be detected.

Until now, micro-events that have been addressed in the Topic Detection and Tracking community were detectable from a single text entry [4–6]. Lastly, no studies were conducted on micro-event detection in the domain of Software Engineering and Stack Overflow Q&A platform dataset in particular.

In our study we consider Free/Libre Open Source Software (FLOSS) version releases as micro-events and Stack Overflow Q&A communications as a platform where associated communities interact.

## Research question

We investigate whether FLOSS version releases can be associated with a characteristic change in the associated community interactions. In other words, we aim at detecting FLOSS version release events from SO message data. We formally define null and alternative hypotheses in the Materials and methods section.

## Contribution

The contributions may be listed as follows:

- We propose a Natural Language Processing (NLP) pipeline for detecting Software Engineering micro-events in noisy forum-like data.

- We perform a feasibility analysis of the approach across a broad range of scenarios. Such elements as a set of considered features (predictors), different estimators, type and length of the detection time window, and type of events are investigated.

- Lastly, we propose a domain-agnostic synthetic data generation model, allowing to generate synthetic forum-like messages with controlled strength of the community response. It helps us to understand scenarios in which the micro-events can be successfully detected.

## Background material and related work

Initially Topic detection and tracking was limited to detecting and following events in news streams. Later the area has benefited from Twitter data [7]. Currently these two streams are being developed in parallel with a certain overlap in methods and approaches [8]. The section is structured as follows: Firstly, we name event attributes which are usually used in event definitions and list event detection settings in terms of machine learning. Secondly, we describe

relevant topic modeling approaches. Thirdly, we discuss common NLP distance metrics. Finally, we link current study to the existing research.

One can distinguish two general types of event detection:

- Retrospective Event Detection (RED)—analysis of the data from the past to discover new, unknown events;

- New Event Detection (NED)—usually done in an online fashion with a stream of data.

The field of event detection is very broad and there are various definitions of the events in the literature. Their classification can be done by the following set of properties:

- Topic—there are topic-specific and general events.

- Geographic location—there are events, relevant for particular locations, like traffic information, concerts, etc. and less location-specific events, like financial reports, online entertainment project releases, etc.

- Time scale—depending on the event, the time horizon might be from hours to weeks and months.

- Reaction—not every event can be associated with a characteristic change in the data source. There might be different reasons for that, and depending on the methodology it may be addressed differently.

The event detection approaches may be also described as Feature Pivot or Document Pivot [8]. The first one involves a certain representation of documents/posts/messages within a time window with changes in the representation indicating the event. The Document Pivot focuses on classification of the documents as related or not related to a particular event.

In terms of machine learning, the detection challenges can be treated as the following:

- Supervised Clustering;

- Semi-supervised Clustering;

- Unsupervised Clustering;

- Classification;

- Anomaly Detection.

## Topic modeling

One of the methods used for detecting events is topic modeling. A well-established technique for that is Latent Dirichlet Allocation (LDA) topic modeling [9], where changes in the topics across time windows might indicate an event. Usually dynamic or temporal implementations of LDA are used [10] for this purpose. However they require certain adjustments, like incremental refitting of the model, and often do not have a scalable implementation available. Gerlach et al. recently proposed a stochastic block model-based method (hSBM) which outperforms LDA [11]. In our study we use both LDA and hSBM approaches to obtain post representations.

Applying LDA topic modeling approach we follow a well-established policy of the model parameters optimization. We feel that it is necessary to relate the existing works using LDA to analyse SO data and the current study. Below, we would like to overview three works—by Barua et al. [12], Yang et al. [13] and Abellatif et al. [14].

First, a study by Barua et al. on analysis of topics and trends in Stack Overflow community [12]. The study applies LDA as a technique for the topic extraction, focusing on empirical analysis of single messages as well as investigation of trends. When assigning topics to posts, the authors use a probability threshold, not considering topics with probabilities below 0.1. While it helps manual post interpretation, ML models might benefit from the less probable topics as these may still contain valid information. The second study is by Yang et al. [13] on analysis of security-related topics in SO data. The authors perform optimization of the number of topics using a genetic algorithm with a Silhouette coefficient as an objective function. The study is focused on topic analysis rather than event detection. Finally, there is a recent study by Abellatif et at. analysing the chatbot community of Stack Overflow [14]. Abellatif et at. manually associate increased posting activity with two technology releases. While it is infeasible to investigate these observations statistically due to a small sample size, we formalize the challenge by defining more subtle events of similar nature, and proposing a dataset with a statistically sound pipeline for investigating them.

## Distance metrics for text data

A core of the NLP methods are distance measures for words, sentences and texts. Jaccard similarity [15], Cosine similarity, Euclidean distance, Averaged Kullback-Leibler (KL) divergence, etc. [16] are commonly used distance measures in Natural Language Processing (NLP). An established example of a more computationally intensive semantic-based approach is Word Mover's Distance (WMD) [17]. When optimizing a synthetic data generator, we use KL divergence measure as a well-known and computationally efficient method of computing distances between distributions. Also, we use Jaccard similarity to naturally compute similarities between messages.

## Related research

A known example of an event detection system in Twitter is TwitInfo proposed by Marcus et al. [18]. The system performs clustering by topic and implies that if activity in a certain topic peaks—an event took place. Then the most relevant tweets are obtained using a keywords-based ranking. However, TwitInfo does not generalize to micro-events due to limitations of the keywords-based ranking.

Another relevant topic is collective event detection, for example flu epidemics. Collective events require multiple posts for detection [19, 20]. At the same time, each post can be successfully labeled as related or not related to the event (like in the study by Aramaki et al. [21]), distinguishing collective events from the introduced notion of micro-events.

Finally, we discuss Multiple instance learning (MIL). It is usually used when the labels of the dataset are partially available. MIL has been used in a range of domains, like object detection [22], audio event detection [23], text categorization [24], etc. In the MIL setting the data are split into subsets. The subset containing no positive instances is labeled as negative and subsets having at least one positive entry—as positive. The classification process usually involves per-instance classification either in a sequential way, using attention [25] and convolution neural networks [26], or in an independent way, using simpler estimators like SVMs [27]. After the per-instance classification there is a pooling procedure, summarizing the output [23]. MIL's approach is similar to the proposed in the current study in considering multiple instances as bags. However, it operates under an assumption that the elementary entries can be labeled, which is not the case in our setting.

## Materials and methods

To support the reliability and reproducibility of the study, we have pre-registered the experiments on the Open Science Framework (https://osf.io/enrd9/?view_only= 0888484923dc46b2b87c90060cb8f961). A pre-registration implies stating upfront the hypothesis, aims, methods and a state of the data collection. While the pre-registrations are relatively uncommon for the field of Computer Science and, especially, Machine Learning, it has been accepted as a good practice for evidence-based methods [28] and commonly used in psychology [29].

The significance level for this study is $\alpha$ = 0.05. The multiple comparisons are accounted for using Holm-Bonferroni corrections. Each type of dataset (Selenium, Django, Multiple) is considered a separate experiment family. We apply the corrections when judging about the significance of the models or model features—each case is explicitly mentioned in the Results section. Also, we have made the data and the code of the study available via Zenodo [30] and Newcastle University data storage [31].

There are two significant issues in the event detection research field—reproducibility and method comparison. The reasons are lack of the datasets standardisation and loose understanding of the notion of an event. While the most commonly used dataset is Twitter, studies are typically restricted to its samples filtered by topic, hashtags, location, language or other rules [32]. We do not compare the obtained results to any other study since there is no study with a comparable definition of the events, however we are making the best effort to ensure reproducibility and transparency of the conducted research.

We illustrate the structure of the study in Fig 1. The current section describes the aims of the project, its sampling strategy, and the methods used for data pre-processing, event detection and model analysis.

### Aim

In order to formalize the research question, the null and alternative hypotheses were formulated.

$H_0$: An event is not associated with a change in the topic distribution or sentiment, representing textual communication of the community.

$H_1$: There is an associated change in the topic distribution or sentiment, representing the textual communication of a community, and the event.

In order to validate the hypotheses, the statistical sample was defined and analysed, as described in the following subsections.

### Dataset description and sampling strategy

We used two data sources in the study: stackoverflow.com and libraries.io. Both of them comply with Attribution-ShareAlike 4.0 International (CC BY-SA 4.0) allowing data sharing and adaption. Older posts of StackOverflow comply with the earlier versions of the license (2.5 and 3.0), still allowing adapting and sharing the data. The datasets were downloaded from the libraries.io and archive.org web-pages, where they were originally shared by the owners of the resources. One every stage of the study we fully comply with the terms and conditions of stackoverflow.com, libraries.io and archive.org web-resources.

The study sample consists of a subset of packages, related to Django web-framework, selected based on the presence of the associated discussions on SO platform and having associated event entries in the libraries.io 1.4.0 dataset [33] (Libraries.io collects, among other

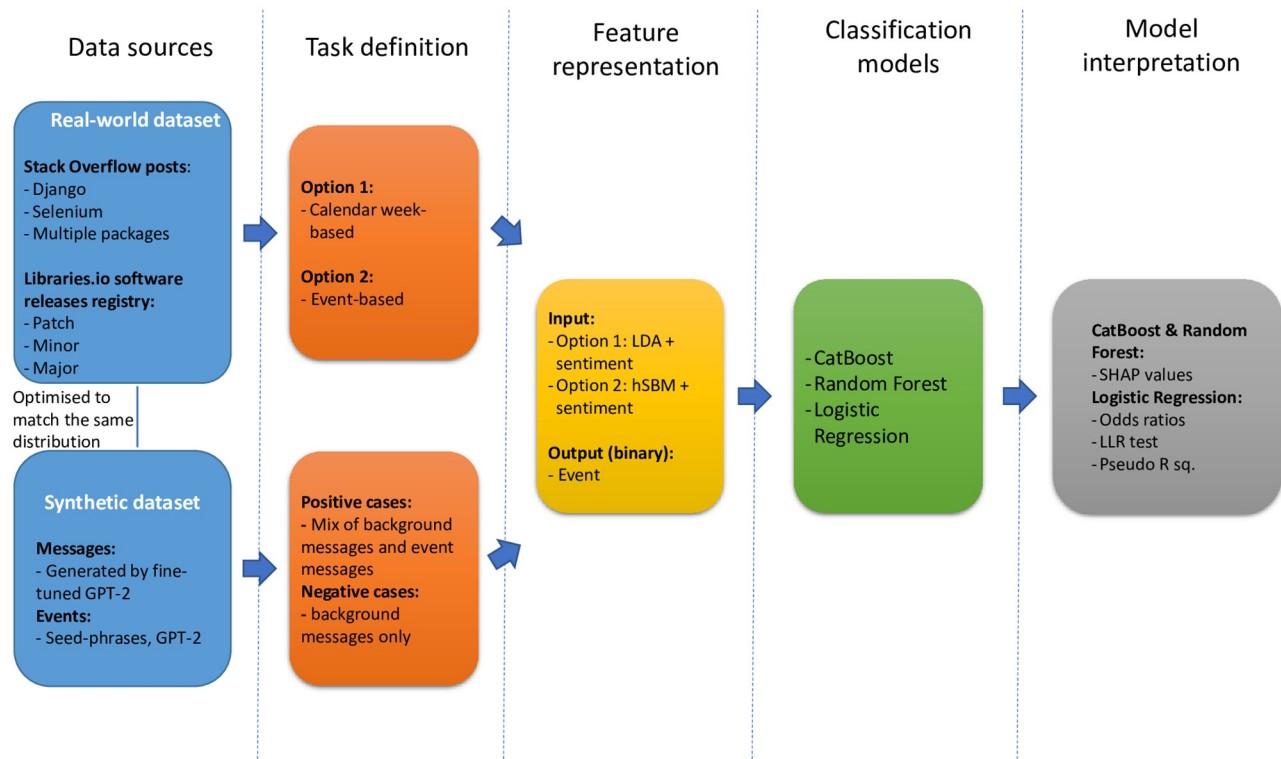

**Fig 1. Principal steps of the study.** The diagram illustrates the study steps of the study. In the synthetic data pipeline the reactions to the event are observed throughout the time step and we do not specify at what point the event takes place.

information, release dates of FLOSS packages). The proposed sample allowed to investigate two dataset configurations—single package SO data sample with associated version releases, and multiple packages.

We manually obtained the initial list of packages from the djangopackages.org. This was done if full compliance with the web-page terms of use. Since, we aimed to ensure the theoretical possibility of event detection for every package, we required the package-associated SO community to be large enough and active. We performed initial filtering of the packages by the number of package followers on Github—all packages with <1000 followers were dropped. On the next step we filtered by the number of SO posts associated with a package—communities with a number of messages < 1% of the number of posts associated with the Django package were dropped. We chose Django as a reference package as it is a well-established package with a large community and package updates. Moreover, Django relies on a number of other packages which might have a positive performance impact on the multiple-package datasets.

We downloaded a complete data dump from Stack Overflow, version June 2018, then we filtered the messages by checking the presence of the selected package names in the body and tags of the messages. We split the dataset into training and test sets by using the first 60% of messages (chronologically) for training and the remaining 40% for test.

Throughout the paper we used the following dataset naming convention: [package][type of event][time step]. We studied the datasets of 3 types: Multiple (the dataset including messages and releases of 7 different packages), Django and Selenium. The event types are major, minor and patch updates. And the time steps are either event-based or calendar week-based (c.w.).

## Dataset design

**Class labels.**  From the list of release dates provided by libraries.io we extracted three types of FLOSS version releases: patch, minor and major updates. When identifying the event types, we applied the Semantic Versioning 2.0.0 convention (https://semver.org/#semantic-versioning-200).

**Datasets.**  We used two largest packages (by community size) to generate the single-package datasets, namely Django and Selenium. Additionally, 7 packages (listed in Fig 2) were used to generate the multiple-packages dataset. We distinguished all three types of updates for the multiple-packages dataset and only patch and minor updates for the single-package datasets. Major updates are too sparse to use them in the single-package datasets. Finally, we have designed two types of time steps.

**Time steps.**  The first design is based on calendar weeks—every calendar week is a single time step. Events take place on any day of the time step (Fig 3(A)). If multiple events take

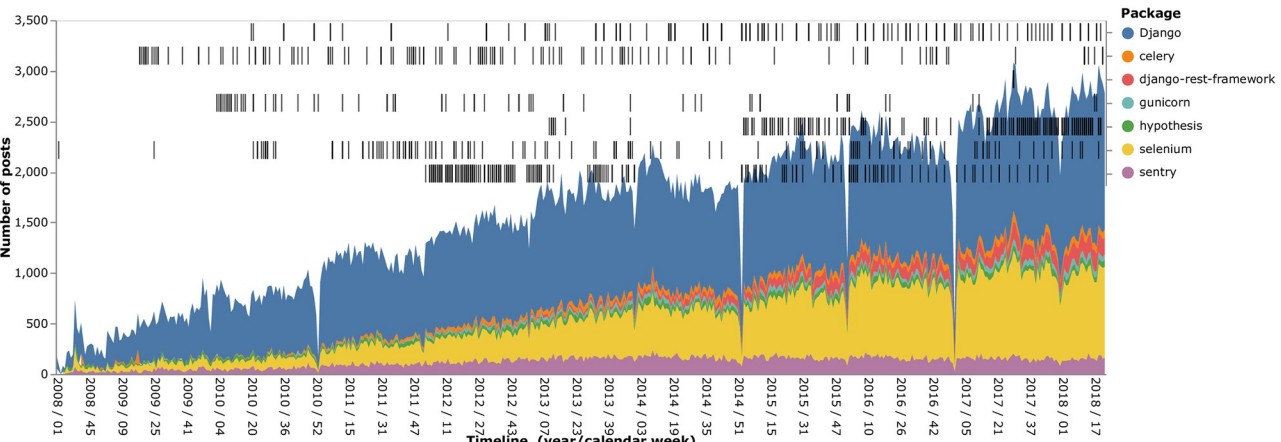

**Fig 2. Number of posts and events used in the study.** Sample of the posts and the events per package for the available time range. The number of posts is provided in a per-week fashion. The packages are stacked vertically. The spike drops take place in the New Year's Eve periods.

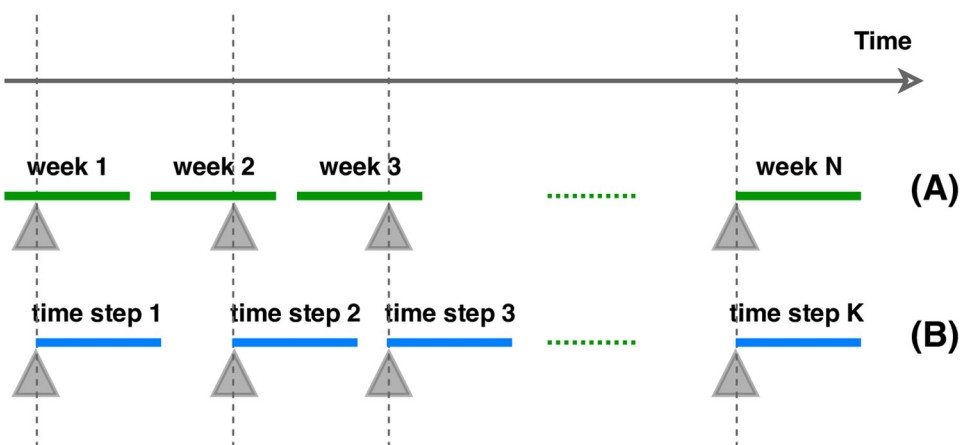

**Fig 3. Time step design.** The figure illustrates two designs of time steps—calendar week-based (A), and event-based (B). Grey triangles indicate events which take place at the beginning of event-based time steps and at any moment of calendar week-based time steps.

place, only the most significant (major>minor>patch) is considered while labeling the time steps. The control time steps are those in which no event of any type took place. This approach is aimed to study prior and posterior characteristic changes in the communication.

The second time step generation is event-based. Each event takes place on the day 0 of the time step (Fig 3(B)). Whenever there are multiple events with less than 7 days of gap, time steps overlap (i.e. the time steps share messages). As control time steps we defined second weeks of 14 day intervals with no event occurring within that period. Because of this restrictive definition messages of certain dates (first weeks of 14 day periods) could not be used for either type of time step and had to be dropped. This design decision minimizes the noise characteristic changes caused by prior events.

The more days a time step consists of, the less entries we get in the dataset. Based on the frequencies on the events and common sense, we decided to set the time step length to 7 days. It allows on average 3.5 days for a developer to relate to the event considering the calendar week granularity of the time steps, and 7 days—for the event-based approach. Daily time steps were studied as well (the experiments are not reported, but available in the code supplement), however, they would require much faster responses from the community and a more prolific communication.

## Pre-processing

Prior to applying the NLP tools, the code snippets and HTML tags were removed from the body of the messages. Initially, five techniques were applied to design the features: sentiment analysis, Bag-of-Words (BoW), LDA topic modeling, hSBM topic modeling and TextRank key phrases extraction. The central analysis of this paper was conducted using two feature spaces: LDA with sentiment analysis and hSBM with sentiment analysis. We feel that these feature spaces allow allow to investigate both small- and large-dimensional experiment setups, and at the same time give the best trade off between the dimensionality and the contained information. The features were extracted on a per-message basis with further grouping by the time step. Other feature spaces were investigated in the preliminary analysis (not reported) and their contribution was found to be limited.

**Sentiment analysis.**   Features were generated by the NLTK python package [34], specifically using the Vader method [35]. There are 4 features generated for each message: negative, neutral, positive and compound components. The latter is a 1d representation of the sentiment. We include sentiment analysis in every feature space, due to its compactness. The sentiment features are coded as "sentiment_<component>", where the components are negative, neutral, positive and compound.

**Latent Dirichlet Allocation (LDA) topic modeling.**   Is a generative model widely used in Natural Language Processing [9]. The model requires Bag-of-Words encoded text as an input and outputs a distribution of topics present in the text. Each topic has a set of associated tokens. In the model training phase the topics are automatically defined based on the word co-occurrences. Concretely: a word subset, occurring across multiple texts (messages, documents, posts, etc.) defines a latent topic. As an example: words "oranges" and "fruits" are seen together more often than "oranges" and "transactions", while "transactions" are often found in the context of "banks". In case of a sufficient amount of documents containing these subsets of words, the model would detect a topic for fruits, represented by "fruits" and "oranges"; and a financial topic, defined by words "banks" and "transactions". An interpretation of topics is usually done manually but the quality of the topics can be assessed in an automatic way.

We optimized and fitted the model on the training part of the dataset and computed fixed-dimensional vectors for posts in the whole dataset. Each dimension represents the probability

of the post belonging to a particular topic. In order to get the per-time step representations, we computed per-topic averages across the messages in the time step.

To apply LDA, we chose the gensim package with a robust and scalable implementation of the method [36]. We built the vocabulary from the lemmatized messages with included bi- and tri-grams. The topic number was optimized on the training dataset. Coherence measure (C_V) was used to find the optimal number of topics. We computed coherence for 3 random-seeded models to ensure the result is not randomness-biased. Finally, we chose the number of topics using Elbow method heuristic.

To verify the soundness of the obtained LDA topics, we manually inspected the top 5 messages associated with each topic, ranked by the gap in the probabilities between the target topic and the next closest one. Finally, showing the general theme of discussions, we assigned names to the LDA topics. The feature names are coded as "lda_topic__<topic name>".

**Hierarchical Stochastic Block Model (hSBM).**   Is a novel approach based on finding communities in complex networks and topic modeling [11]. Concretely: it represents the dataset as a bipartite graph of posts and words. The model outperforms LDA leading to better topic models. Also, hSBM does not require setting the number of topics allowing the model to find them naturally. We used this method to obtain a larger feature space and ensure that our feasibility analysis covers the dimensionality aspect. We computed the per-time step representations by taking topic means across posts in the time step, the same way as for LDA.

**Other features.**   In exploratory analysis we also evaluated the performance of the TextRank and Bag-of-Words NLP representations. However, using these did not improve the performance of the models. There are multiple reasons of not including these features into the main analysis:

- Their dimensionality is around 2 orders larger in comparison to the used approaches, making the models susceptible to the curse of dimensionality. [37];

- We assume that lower-level features, such as BoW, contribute to the output as subsets. Of course, discovering feature interactions in the setting where the number of features is orders larger than the number of entries is bound to detect spurious interactions and, consequently, leads to interpretation errors.

Lastly, we investigated a more complex per-time step feature design, where each topic was represented by a minimum, maximum and mean values together with the size of each topic computed as difference between maximal and minimal values (the experiments are not reported, but present in supplementary code). It did not lead to any significant improvement of the models.

**Standardization.**   As we wanted to make the features general across time steps, we standardized them as the following:

$$z = \frac{x_i - \mu}{\sigma}, \tag{1}$$

where $\mu$ is the sample average and $\sigma$ is the sample standard deviation. Both values were obtained from the training partition of the data.

## Analysis pipelines

We perform the data analysis using three different estimators—Logistic Regression (LR), Random Forest (RF) and CatBoost (CB) [38]. We selected those in order to cover a rigorous statistical approach (LR), well-known machine learning approach (RF) and a robust state-of-the-art machine learning approach (CB). For the Logistic Regression, we perform a full stack of model

investigation techniques recommended by Field et al. [39] to ensure that the obtained models are statistically reliable. Both Random Forest and CatBoost are based on decision trees and there is a well-established set of direct model interpretation tools, which we apply in the current study.

We perform the same set of steps for all the estimators in terms of feature selection, model optimization, fitting and performance assessment. We ensure as uniform pipeline design as possible between the three estimators. However, the model analysis differs since the approaches are model-specific.

Due to the limitations of the hSBM implementation, when analysing the synthetic data, we limit ourselves to the LDA-based feature space. Concretely: RAM requirements, lack of scalability and robustness of the existing implementation make it infeasible to obtain hSBM topics for synthetic data.

**Description of the Logistic Regression pipeline.**    *Effect sizes.* We computed effect sizes to get a model-agnostic quantitative measure of the phenomenon magnitude. The benefit of the method is an ability to directly compare strengths of different phenomena irrespective of the models used in the study. We used Cliff's Delta [40] as an effect size measure, due to a non-normal feature values distribution. R effsize package was used to compute the effect sizes. Additionally, we computed 0.95 confidence intervals. Finally, we corrected 0.95 confidence intervals for multiple features.

*Outliers.* Prior to fitting LR models, we capped outliers using the Tukey method [41], as they are known to unduly affect the Logistic Regression model results. We used R *boxplot.stats* () implementation for this purpose. We obtained the capping thresholds from the training partition of the data and capped the whole dataset.

*Feature selection.* We performed feature selection using recursive feature elimination with cross-validation (RFECV). This method is a commonly adopted standard for feature selection in the machine learning community.

The method works as the following: the model is fitted with the full set of features, cross-validated, then the least important feature is dropped, the model is fitted again. This is repeated until there is a single feature in the model. The best subset is chosen based on the cross-validated performance of the model. As a performance metric we used precision-recall area under curve (PR-AUC), which is commonly used for imbalanced datasets. A fraction of features can be removed on each iteration to speed-up the process.

*Parameter tuning and model fitting.* The only tunable parameter in the *R glm* (generalized linear model) implementation of LR is class weights. However, uneven class weights affect the base probability of the model as well as skew the output distribution. Consequently, it decreases interpretability of the model and makes any finding inaccurate and inapplicable to the population. We decided to work with even class weights to avoid the mentioned issues.

Log-Likelihood Ratio (LLR) test was performed to check the significance of the model fit for the optimal feature space. We find it important to note that no test data is used to conduct the test and it does not say anything about the model performance.

*Model analysis.* The assessment of the model quality involved the following procedures:

1. We estimate goodness of fit using Tjur, Nagelkerke, Cox-Snell, and Adjusted McFadden pseudo $R^2$ measures.

2. We computed odds ratios of the model and plotted them as a forest plot with the 0.95 confidence intervals. The intervals were corrected for multiple comparisons. Since the data were standardized, one standard deviation change in the feature value leads to the odds ratio multiplicative change in the base probability: for the base probability $p$ and the odds ratio

value $d$, a standard deviation feature value change results in the updated probability of $p \times d$.

3. To assess the model quality, we computed Variance Inflation Factors (VIFs). They allow spotting potentially redundant features in the model. As advised by Andy Field et al. [39], VIFs exceeding 10 indicate unreliability of the model.

4. Since logistic regression operates under assumption of linear relations between inputs and outputs, we verified whether it holds by adding $log(f) \times f$ features to the model and checking their significance [39].

5. To make sure there are no influential outliers in the fitted model, we applied Bonferroni Outlier Test on the fitted model. It allows spotting influential cases significantly changing the behaviour of the model. Additionally, added variable plots were built and visually assessed (not reported, present in the supplementary code).

*Model performance*. Since presence of the event is minority in most datasets and majority in Multiple packages dataset, we computed a mean of PR-AUC, treating events as a positive and a negative class. This metric is used across the study. Additionally, we use PR-AUC as a performance metric in permutation tests. Lastly, we provide alternative metrics in S2 Appendix to allow better understanding of the results.

**Description of the Random Forest and CatBoost pipelines.** *Feature selection*. Recursive feature elimination with cross-validation (RFECV, *sklearn* implementation with 2-fold time series split) feature selection requires a classifier to provide a feature ranking. While for the LDA feature space, the features we removed features one-by-one, hSBM feature space is larger and we choose to drop 10% of bottom-ranked features to make the experiments computationally feasible. At this point we used the default estimator parameters, as both RF and CB as known to be rather unpretentious in terms of parameter tuning.

*Parameter tuning*. To tune the model parameters we used a Grid Search approach with a 2-folded time series split cross-validation. We optimized the number of trees, maximal tree depth and class weights. For CatBoost we additionally optimized tree L2-regularization and a use of the temporal dimension (binary variable).

*Model analysis*. For RF and CB estimators we generated SHAP values and visualized feature impacts on a per-entry basis using shap python package implementation [42]. Taking into account feature values and their impact, SHAP provides theory-grounded and more reliable feature importance estimates in comparison to the built-in sklearn and catboost packages feature importance.

*Model performance*. For CB and RF we compute the same set of performance metrics as for LR.

## Synthetic data generation

We generated a synthetic dataset with an objective of understanding, what strength of the reactions in the textual communications can be reliably detected.

The section is structured as the following: i) We briefly describe the data generation process; ii) We go into detail in the Deep learning text generator subsection.

The sequence of the synthetic data study phases is shown in Fig 4.

When generating the synthetic data, we separately generate control and event-related messages. After the messages are generated, we form time steps by bagging messages of two types in a variable proportion. To assess the model performance we follow the same feature design procedure as in case of SO data.

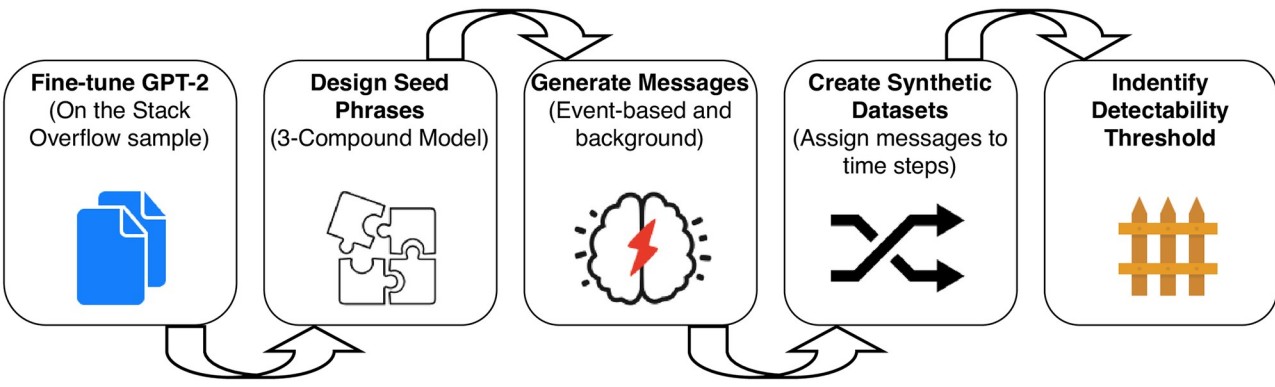

**Fig 4. Sequence of steps for the synthetic dataset.**

The process of data generation can be seen as a set of steps:

1. Generate background messages;

2. Generate event-related messages;

3. By bagging messages from (1) and (2) we create time steps with positive labels;

4. We bag messages from (1) to create negatively labeled time steps;

We assumed that in the real world scenario only a fraction of messages is relevant to an event—by changing the proportion of the two message types in step 3) we investigate this assumption.

**Synthetic data analysis pipeline.** After generating the data, we wanted to find the detectability threshold for the micro-events. We define the threshold as a minimal fraction of event-related messages, for which micro-events can be detected with a significant outcome of the permutation test.

We perform the same feature selection, optimization, and performance assessment for the SO datasets and synthetic data.

Additionally, to reduce the influence of randomness, we repeat the experiments multiple times for each dataset configuration by reshuffling messages across the time steps.

**Deep learning text generator.** The posts were generated using a small version of GPT-2 OpenAI neural network [43] with 117M neurons. The original pre-trained version of GPT-2 was fine-tuned on the multiple packages sample of Stack Overflow data.

*Message types*. The background or control messages, having no relation to an event, were generated from an empty context. In order to generate event-related messages, we propose a model to inject events into the data. The community values are represented by 3 compounds: rules, people and products (Fig 5). Changes in any of these compounds might lead to changes in the community communications. These are abstract entities, specific for every community.

We generated event-related messages from a set of seed phrases, used as a context for the generator network.

The seed phrases are designed on the basis of the real-world dataset, used for the generator fine-tuning. Entities representing the three compounds (rules, people and products) have to be identified in the Stack Overflow dataset. For that purpose we used the Word2Vec model [44], fitted on Google News dataset. We expect this model to be generic and cover a wide range of topics including IT.

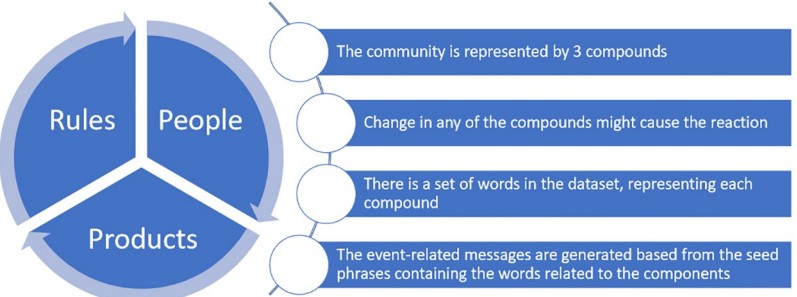

**Fig 5. Three compound model.** It is used for the event-related messages generation. It assumes that the textual representation of the community in the forum-like platforms consists of 3 compounds and any change in them (event) might lead to the reaction. This model is used to generate the event-related messages.

We obtained the closest (cosine similarity) 1.7k nouns for each of the components from SO dataset. Since events are represented by changes in the compounds, we took 1k closest verbs representing addition and removal operations.

Finally, we obtained the seed phrases for generating the event-related messages by concatenating the selected nouns and verbs. For the generation we used a random set of 100 seed phrases. We used NLTK POS tagger for the Part of Speech identification.

We provide a detailed description of the generator optimization and ways of validating the synthetic data quality in S3 Appendix.

## Deviations from the pre-registration

After the first tests with the multiple packages dataset, we decided to simplify the task and run the classification on the single package datasets. Additionally, we decided to use a statistical pipeline to better understand the challenge. Finally, we designed a synthetic data generation method and successfully used the synthetic data to find the micro-event detectability threshold. We have preserved the pre-registered model performance assessment measures and tests and applied them uniformly to all the experiments.

## Results

### Data sample

**Stack Overflow data.** In the single package experiments we focused on Django and Selenium, as they account for around 85% of the messages. Table 1 shows the packages (and

**Table 1. Numbers of posts per package.**

| Package | Number of posts |
|---|---:|
| Django | 475760 |
| Selenium | 210498 |
| Sentry | 60874 |
| Django-rest-framework | 24852 |
| celery | 20864 |
| Hypothesis | 19401 |
| Gunicorn | 14602 |
| **Total (unique)** | **826851 (777812)** |

Number of posts per package after filtering by the package name in the post's tag or body. The total number of posts in the Stack Overflow platform data dump of 06/2018 is **65049182**.

number of associated messages) included in our sample. Moreover, Fig 2 represents chronologically the events associated with these seven packages. Black strips at the top part of the figure represent the events. We provide this figure to support the reproducibility of the study.

The posts before 27 July, 2015 belong to the training set, after the date—to test.

**Synthetic data.**   The optimized generator was used to obtain 1109150 background messages and 154680 event-related messages.

We generated 15 instances of each dataset configuration with the event-related message fractions in a range from 0.1 to 0.45 with a step of 0.05. Each synthetic dataset consists of 335 time steps, to match the number of time steps in the SO dataset, ranging from 171 to 708 time steps. The ratio of positive to negative time steps was set to 0.25 to mirror the considered real-world scenarios (mean event fraction in the SO datasets is 0.26).

The details on how the messages and the time steps were generated are provided in the Synthetic data generation section.

## SO datasets results

**Summary of the results.**   We list the performance summary of the three estimators (Table 2) obtained by fitting and tuning on the training dataset batch and assessing performance on the test batch of the dataset. We provide more metrics in S2 Appendix. Feature selection leads to larger feature spaces in case of CB and RF estimators—this is observed for the hBSM feature space (Table 2 of S2 Appendix), it is less obvious from LDA feature space results (Table 1 of S2 Appendix). Once the corrections for multiple comparisons are applied within each experiment family, the permutation tests reveal 7 significant models in total. Interestingly, 6 of them belong to the multiple packages dataset. Considering the PR-AUC metric, we see that models' performance in most cases only marginally better than the baseline of 0.5.

To further analyse results and understand the limitations of the models in this section we focus on a particular dataset as a case study: Selenium package, patch updates, event-based time steps dataset. We chose this dataset based on the goodness of fit of the LDA features space. Concretely: we assessed Adjusted McFadden and Tjur $R^2$ measures (see S1 Appendix). We did not consider multiple packages datasets with event-based time steps due to their violation of the independence assumption of the logistic regression—a message may be included in multiple time steps simultaneously, as described in the Dataset design section. Also, when considering the goodness of fit, we restrict ourselves to LDA feature space due to artificial behaviour of some metrics caused by large dimensionality of hSBM feature space.

**Effect sizes.**   As a model-agnostic feature analysis, we build a forest plot of the effect sizes, sorted by the range of the confidence intervals (CI) (Fig 6).

The error bars illustrate the 0.95 CIs allowing to assess significance of the features beyond the considered sample. There are 18 features in total—14 LDA topics and 4 sentiment-related ones. At this point, there are only 3 features with significant effect sizes—Servers, Testing and Web Elements LDA topics.

We do not report effect sizes of hSBM feature space due to its dimensionality leading to large CIs and, consequently, insignificance of all the features.

**Logistic regression analysis.**   *Model fitting analysis.* We have fitted the logistic regression model (Tables 3 and 4) and computed their goodness of fit measures. After the feature selection step, there are 13 features in case of LDA feature space. After correcting for multiple comparisons, 5 of them are significant—Intercept, Template Tags, Testing, Forms and Models and Servers LDA topics. Based on Tjur $R^2$ one can see that around 16% of the variance is explained, other $R^2$ measures support that statement. It should be noted that pseudo $R^2$ typical values tend to be smaller than linear regression $R^2$. Concretely: McFadden suggests

**Table 2. CB, RF and LR model performance.**

| | hSBM feature space | | | | | | | | |
|---|---|---|---|---|---|---|---|---|---|
| Dataset | CatBoost | | | RF | | | LR | | |
| | PRAUC | P.test | F1-score | PRAUC | P.test | F1-score | PRAUC | P.test | F1-score |
| Multiple major event-based | 0.54 | 1.00 | 0.16 | 0.52 | 1.00 | 0.16 | 0.52 | 0.76 | 0.20 |
| Multiple minor event-based | 0.50 | 0.01 | 0.51 | 0.75 | <0.001* | 0.49 | 0.50 | 0.90 | 0.36 |
| Multiple patch event-based | 0.50 | 0.81 | 0.35 | 0.50 | 0.88 | 0.41 | 0.51 | <0.001* | 0.49 |
| Django minor event-based | 0.50 | 0.26 | 0.50 | 0.55 | <0.001* | 0.47 | 0.51 | 0.01 | 0.41 |
| Django patch event-based | 0.52 | 1.00 | 0.46 | 0.54 | 1.00 | 0.47 | 0.51 | 0.15 | 0.42 |
| Selenium minor event-based | 0.52 | 1.00 | 0.46 | 0.50 | 1.00 | 0.46 | 0.50 | 1.00 | 0.46 |
| Selenium patch event-based | 0.50 | 1.00 | 0.44 | 0.51 | 1.00 | 0.44 | 0.50 | 1.00 | 0.44 |
| Multiple major c.w.-based | 0.50 | 1.00 | 0.46 | 0.50 | 1.00 | 0.46 | 0.51 | 1.00 | 0.47 |
| Multiple minor c.w.-based | 0.50 | 0.03 | 0.48 | 0.51 | 0.29 | 0.45 | 0.51 | 0.01 | 0.53 |
| Multiple patch c.w.-based | 0.51 | 0.14 | 0.51 | 0.51 | <0.001* | 0.40 | 0.51 | 0.02 | 0.46 |
| Django minor c.w.-based | 0.51 | 0.04 | 0.46 | 0.51 | 1.00 | 0.45 | 0.50 | 0.54 | 0.45 |
| Django patch c.w.-based | 0.50 | 1.00 | 0.48 | 0.50 | 1.00 | 0.48 | 0.50 | 1.00 | 0.48 |
| Selenium minor c.w.-based | 0.51 | 1.00 | 0.47 | 0.56 | 0.07 | 0.35 | 0.51 | 0.10 | 0.47 |
| Selenium patch c.w.-based | 0.52 | 0.44 | 0.45 | 0.52 | 1.00 | 0.47 | 0.50 | 0.08 | 0.54 |
| | LDA feature space | | | | | | | | |
| Dataset | CatBoost | | | RF | | | LR | | |
| | PRAUC | P.test | F1-score | PRAUC | P.test | F1-score | PRAUC | P.test | F1-score |
| Multiple major event-based | 0.56 | 0.07 | 0.58 | 0.57 | 0.06 | 0.58 | 0.40 | 0.88 | 0.49 |
| Multiple minor event-based | 0.51 | <0.001* | 0.45 | 0.59 | 0.19 | 0.52 | 0.45 | 0.93 | 0.45 |
| Multiple patch event-based | 0.51 | <0.001* | 0.46 | 0.70 | 0.01 | 0.51 | 0.50 | 0.98 | 0.46 |
| Django minor event-based | 0.51 | 0.50 | 0.46 | 0.53 | 0.02 | 0.44 | 0.51 | 0.36 | 0.43 |
| Django patch event-based | 0.54 | 0.15 | 0.55 | 0.50 | 0.30 | 0.51 | 0.47 | 0.44 | 0.47 |
| Selenium minor event-based | 0.51 | 1.00 | 0.45 | 0.51 | 1.00 | 0.47 | 0.46 | 0.98 | 0.47 |
| Selenium patch event-based | 0.50 | 0.81 | 0.46 | 0.50 | 0.03 | 0.46 | 0.52 | 0.46 | 0.45 |
| Multiple major c.w.-based | 0.53 | 0.18 | 0.54 | 0.50 | 0.23 | 0.50 | 0.48 | 0.80 | 0.48 |
| Multiple minor c.w.-based | 0.51 | <0.001* | 0.37 | 0.51 | 0.22 | 0.50 | 0.54 | 0.19 | 0.52 |
| Multiple patch c.w.-based | 0.50 | 0.40 | 0.52 | 0.51 | 0.49 | 0.46 | 0.46 | 0.89 | 0.32 |
| Django minor c.w.-based | 0.50 | 1.00 | 0.44 | 0.50 | 1.00 | 0.44 | 0.50 | 0.52 | 0.44 |
| Django patch c.w.-based | 0.50 | 0.21 | 0.53 | 0.51 | 1.00 | 0.44 | 0.48 | 0.86 | 0.47 |
| Selenium minor c.w.-based | 0.50 | 1.00 | 0.45 | 0.51 | 1.00 | 0.48 | 0.48 | 0.72 | 0.48 |
| Selenium patch c.w.-based | 0.51 | 1.00 | 0.46 | 0.51 | 1.00 | 0.46 | 0.52 | 0.21 | 0.47 |

We evaluate the three estimators on all the datasets. PR-AUC and f1-score metrics are computed as averages of events being positive and a negative class. P.test columns contain p-values of the permutation tests over 1000 permutations. After applying Holm-Bonferroni corrections the threshold for the top 1 model is 0.0014. The significant entries are marked with a star (*). We report more metrics in S2 Appendix.

that McFadden pseudo $R^2$ values of 0.2-0.4 correspond to an excellent model fit [45]. Overall, we see that the pseudo $R^2$ values do not show any anomalies. Correcting for multiple comparisons, Log-Likelihood test outcome indicates that the model fit is significant. hSBM feature spaces model contains only two features including intercept. Even though the topic feature is significant, $R^2$ and LLR test metrics indicate a poor model quality and close to no variance explained (Table 4).

We compute odds ratios of the models (Figs 7 and 8) to assess per-feature contributions to the model output. The error bars correspond to 0.95 CIs. Since the data are standardized, one standard deviation change in the feature value leads to the event probability change

## Feature effect sizes for minor updates

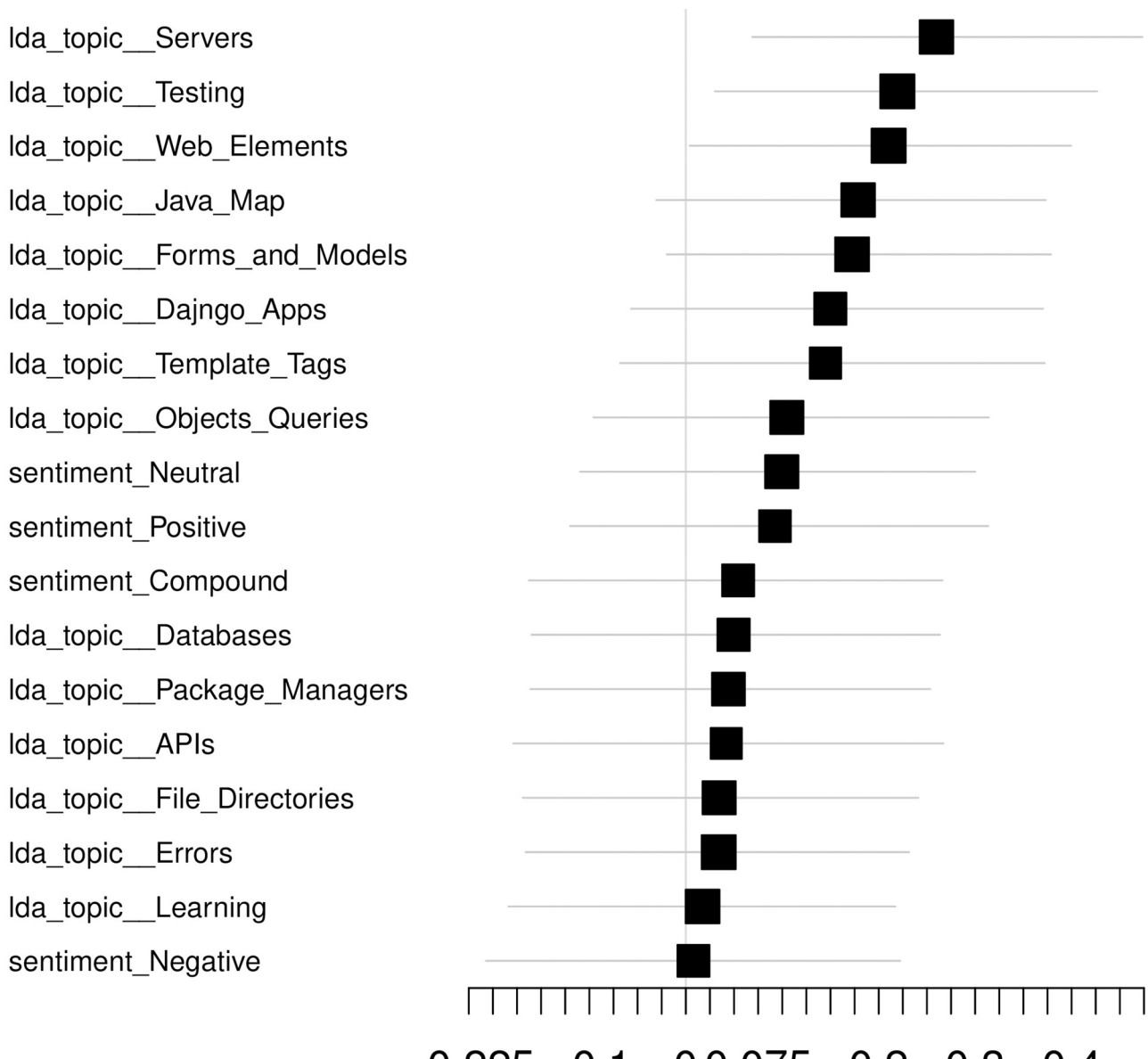

**Fig 6. Dataset effect sizes.** Effect sizes computed for the Selenium package, minor updates, event-based time steps dataset, LDA feature space. Cliff's Delta is used as the effect size measure to account for the non-normal distribution of the feature values. The error bars account for 0.95 confidence intervals (CIs). Features, whose lower CI bound is greater than 0 are considered to be statistically significant. Interpreting the CIs—there is a 0.95 probability that the effect size computed on the population is in the bounds of the CI.

multiplicative of the feature's odds ratio. Significant features are ones whose CIs do not cross a vertical line at $X = 1.0$.

Let us interpret the feature impact on the model output on the example of LDA feature space. If considering significant features with odds ratios around 2 and the model base probability equals 0.22, one standard deviation increment in any of the features leads to the event probability increasing twice—to 0.44. Due to intercept odds ratios being far from neutrality (1.0), positive model output might require multiple features activated.

**Table 3. Logistic regression model, LDA.**

| Predictor | Estimate | Std. Error | Z-value | Pr($>$ \|z\|) | VIF |
|---|---|---|---|---|---|
| (Intercept) | -1.68 | 0.19 | -8.94 | $< 0.001^*$ | |
| lda_topic__Web_Elements | 0.44 | 0.33 | 1.34 | 0.179 | 3.93 |
| lda_topic__Package_Managers | 0.28 | 0.18 | 1.58 | 0.114 | 1.15 |
| lda_topic__Template_Tags | 0.58 | 0.18 | 3.13 | $0.002^*$ | 1.35 |
| lda_topic__Testing | 1.14 | 0.36 | 3.19 | $0.001^*$ | 5.37 |
| lda_topic__Dajngo_Apps | 0.38 | 0.18 | 2.17 | 0.03 | 1.22 |
| lda_topic__Errors | 0.41 | 0.22 | 1.82 | 0.068 | 1.75 |
| lda_topic__Forms_and_Models | 0.57 | 0.18 | 3.11 | $0.002^*$ | 1.24 |
| lda_topic__Servers | 0.57 | 0.19 | 3.07 | $0.002^*$ | 1.44 |
| lda_topic__File_Directories | 0.31 | 0.19 | 1.65 | 0.099 | 1.27 |
| lda_topic__Objects_Queries | 0.36 | 0.18 | 1.96 | 0.05 | 1.18 |
| sentiment_Positive | 0.58 | 0.24 | 2.42 | 0.015 | 2.03 |
| sentiment_Compound | -0.49 | 0.25 | -1.98 | 0.048 | 2.19 |
| **Fit Measurements** | | | | | |
| LLR Test Chi$^2$ | 50.8 | Observations | | 329 | |
| Log Likelihood | -147 | Null model Log Likelihood | | -173 | |
| LLR Test p-value | $<.001$ | Degrees of freedom | | 13 | |
| AIC | 321 | Adj. McFadden $R^2$ | | 0.07 | |
| Null model base probability | 0.22 | Cox-Snell $R^2$ | | 0.14 | |
| | | Nagelkerke $R^2$ | | 0.22 | |
| | | Tjur $R^2$ | | 0.16 | |

LR model fit with its assessment of the goodness of fit. The model was fitted on Selenium package, event-based time steps, minor updates dataset, LDA feature space, with the update events as a dependent variable. The subset of features was selected using the RFECV method with a step of 1. After applying Holm-Bonferroni corrections, significant p-values are marked with a start (*).

*Validating model assumptions*. Logistic regression algorithm assumes there is a linear relation between features and the output. We assess extending the feature space with interaction terms and checking their significance. For the considered dataset configuration, LDA feature space, we find that Errors LDA topic ($F \times log(F)$) is significant—the linearity criterion is

**Table 4. Logistic regression model, hSBM.**

| Predictor | Estimate | Std. Error | Z-value | Pr($>$\|z\|) | VIF |
|---|---|---|---|---|---|
| (Intercept) | -1.29 | 0.13 | -9.54 | $<.001^*$ | - |
| hsbm_topic_310 | 0.24 | 0.12 | 1.98 | $0.048^*$ | - |
| **Fit Measurements** | | | | | |
| LLR Test Chi$^2$ | 3.92 | Observations | | 329 | |
| Log Likelihood | -172.87 | Null model Log Likelihood | | -171 | |
| LLR Test p-value | 0.048 | Degrees of freedom | | 1 | |
| AIC | 346 | Adj. McFadden $R^2$ | | 0.0 | |
| Null model base probability | 0.22 | Cox-Snell $R^2$ | | 0.01 | |
| | | Nagelkerke $R^2$ | | 0.02 | |
| | | Tjur $R^2$ | | 0.01 | |

LR model fit with its assessment of the fit quality. The model was fitted on Selenium package, event-based time steps, minor updates dataset, hSBM feature space, with the update events as a dependent variable. The subset of features was selected using the RFECV method, where 10% of features were removed per iteration. After applying Holm-Bonferroni corrections, the significant p-values are marked with a start (*).

## Odds Ratios for minor updates

**Fig 7. Odds ratios, LDA.** Computed for the model fitted on Selenium package, minor update events, event-based dataset, LDA feature space. X axis is logarithmic and the features are sorted by the confidence interval range.

violated. We observe this for the hSBM feature space as well—non-linear feature of topic 310 is also significant. We conclude that there is a non-linearity present in both cases and they cannot be detected using the LR model. RF and CB do not have any constraints on the data linearity and are capable of detecting the effects missed by LR.

None of the Variance Inflation Factors (VIFs) cross the threshold of 10 (Table 3), meaning there are no superfluous features in the model. Since there is only one topic feature in the hSBM case, no VIF is computed. Bonferroni Outlier Test did not discover any significant outliers, meaning that there are no influential cases in the training data.

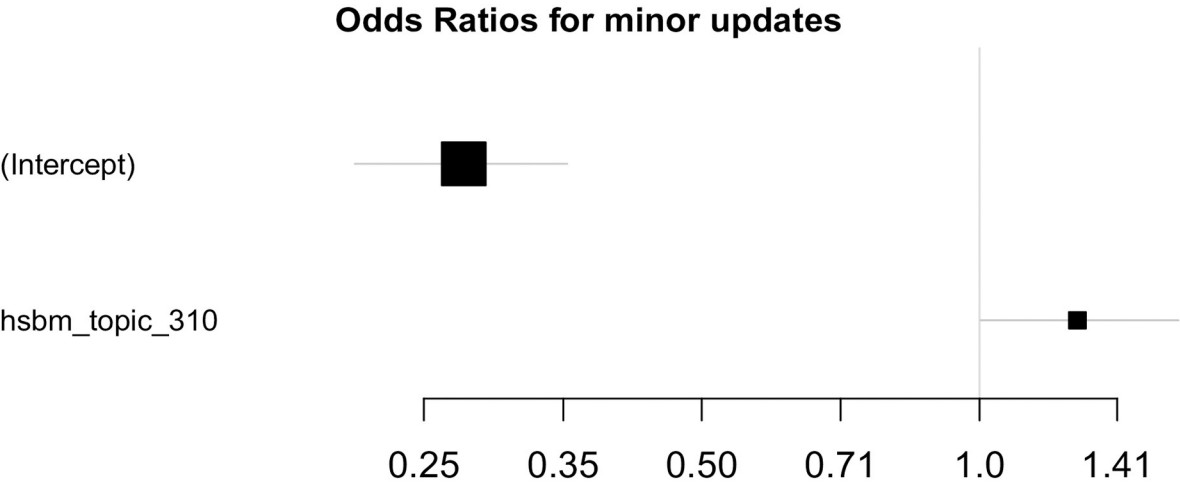

**Fig 8. Odds ratios, hSBM.** Computed for the model fitted on Selenium package, minor update events, event-based dataset, hSBM feature space. X axis is logarithmic and the features are sorted by the confidence interval range.

**Random Forest and CatBoost analysis.** We tuned model parameters using a grid search approach with time series cross-validation. We report the optimized parameters in Table 5.

We plot SHAP values with respect to the positive class output for RF and CB, both feature spaces in Figs 9–12.

The optimized LDA feature space consists of 6 and 4 features for RF and CB, respectively. More often smaller feature values contribute towards a negative class output. Also, there are 3 common features in the models. Moreover, they affect the output in the same way. For most of the features the maximum impact towards a negative output is stronger than maximum impact towards positive—this is shown by larger absolute SHAP values to the left from zero value (X axis). This leads to a more probable negative class output of the model (agrees with the LR model). When we obtain a confusion matrix, the majority class is predicted for all entries in both models. From the odds ratios of LR, and SHAP values, one can see that the Testing topic has the largest impact on the model output from all the LDA topics for all three estimators. The nature of impact is similar as well—larger feature values contribute towards a positive output. Forms and Models topic (which is significant in the LR model) is also present in all the models. Its impact is shifted for CB and RF—there are a number of entries where middle to large values of the feature cause no impact or contribute towards a negative class output.

**Table 5. Optimized parameters of Random Forest and CatBoost models.**

| | hSBM feature space | | LDA feature space | |
|---|---|---|---|---|
| | **Random Forest** | **CatBoost** | **Random Forest** | **CatBoost** |
| Depth | 8 | 6 | 8 | 6 |
| Trees number | 200 | 200 | 50 | 500 |
| Class balance | s-balanced | balanced | balanced | balanced |
| L2 Regularization | - | 3 | - | 7 |
| Temporal nature | - | True | - | True |

The table shows the final parameters of Random Forest and CatBoost estimators for the Selenium package, minor updates, event-based time step dataset. "s-balanced" value accounts for "subsample-balanced" type of class label balancing. Depth corresponds to the maximum depth of a decision tree. Temporal nature is a binary variable, indicating whether the temporal nature of data should be should be considered when fitting the model. The last two parameters were optimized only for CatBoost.

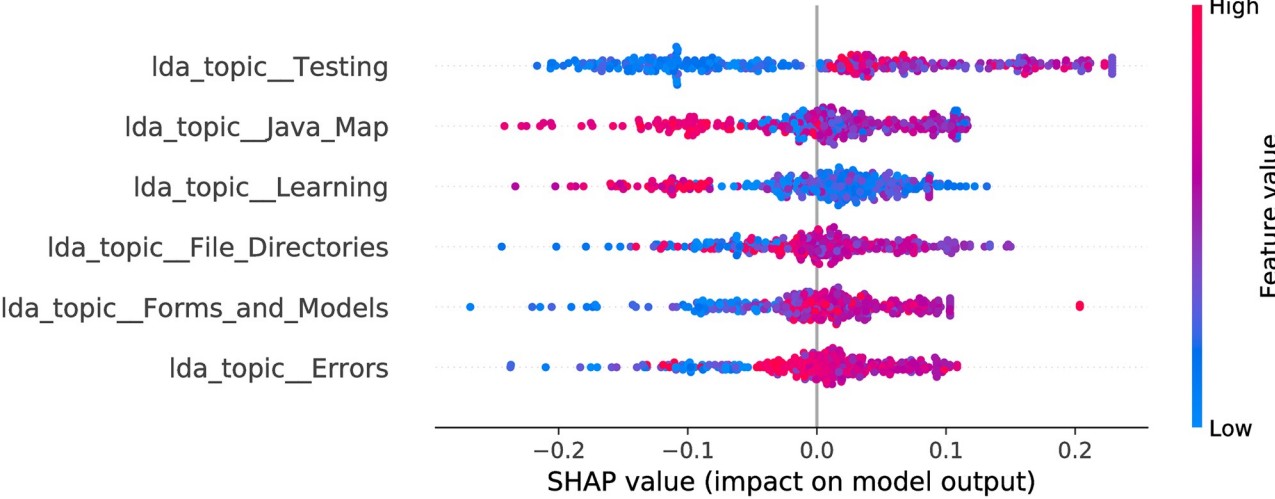

**Fig 9. Random Forest model SHAP values, LDA feature space.** The figure illustrates the influence of the features on the output of the model fitted on Selenium package, minor updates, event-based time steps dataset. Color encodes the feature value and the X axis represents the impact of the feature in a particular case. The RF features partially overlap with the LR—there are 4 common features, 2 of which are significant in the LR. There are 3 features overlapping with the CB model.

Considering hSBM feature space, the optimized feature subsets are large, hence in Figs 11 and 12 we show only top 10 features for each model. Due to the large number of hSBM topics it is not feasible to interpret all of them. Consequently, later in the paper we interpret top 3 impactful features of each model. Overall, there are 441 features overlapping between RF and CB models.

**LDA topics interpretation.** LDA was optimized on the training dataset and 14 topics setting has the optimal coherence C_V score based on Elbow heuristic, as shown in Fig 13.

In this study we interpret in detail significant features of the LR model and all the features of RF.

**File Directories** LDA topic—the topic contains messages on the operations with files and file systems, like accessing, storing and downloading. The top 4 characteristic words extracted from the LDA model are: 'file', 'image', 'directory', 'folder'.

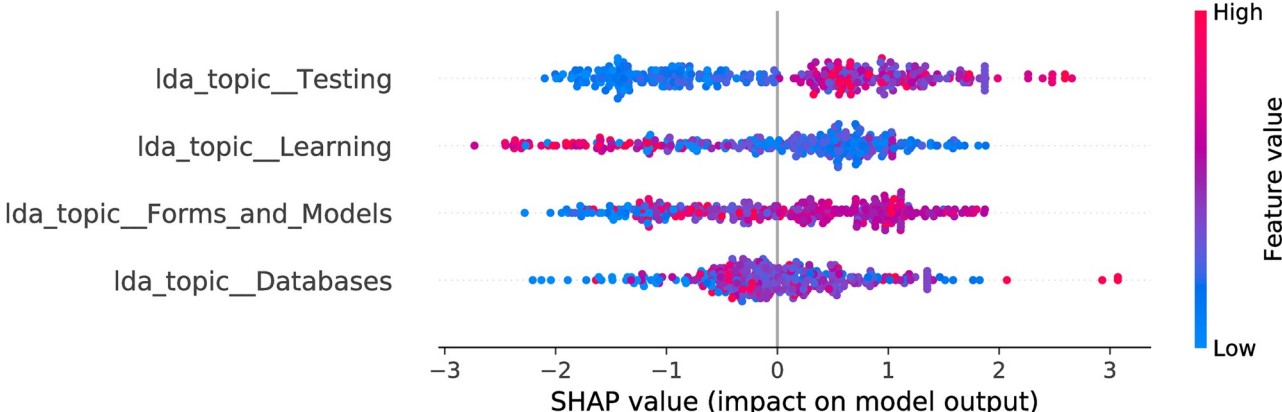

**Fig 10. CatBoost model SHAP values, LDA feature space.** The figure illustrates the influence of the features on the output of the model fitted on Selenium package, minor updates, event-based time steps dataset. Color encodes the feature value and the X axis represents the impact of the feature in a particular case. There are two features which overlap with the LR model and 3 features overlapping with the RF model.

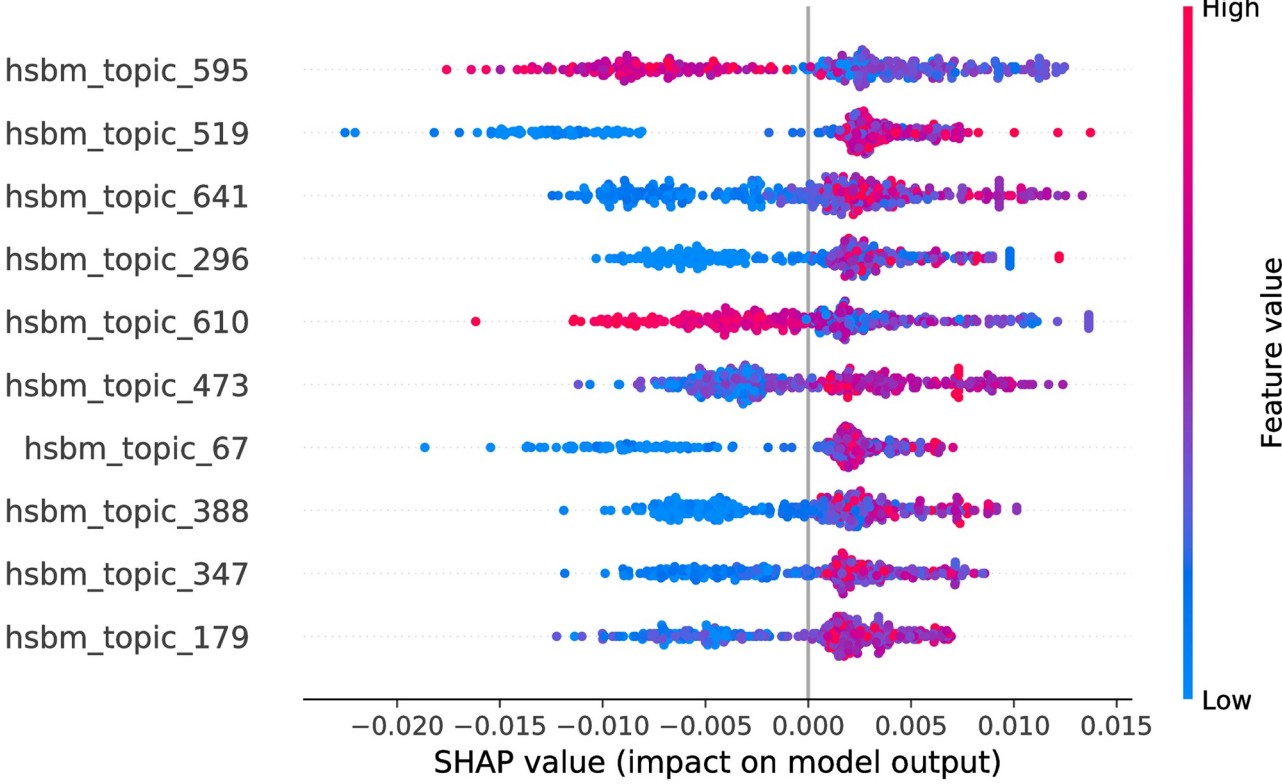

**Fig 11. Random Forest model SHAP values, hSBM feature space.** The figure illustrates the influence of the features on the output of the model fitted on Selenium package, minor updates, event-based time steps dataset. Color encodes the feature value and the X axis represents the impact of the feature in a particular case. Only top 10 features from the feature space are shown.

**Template Tags** LDA topic—communication on Django template language, use of tags in the context of this language. The tokens are 'template', 'view', 'use', 'url'.

**Forms and Models** topic—posts on Django models and forms. These two notions are related in a way that models provide access to databases and forms are used to input the data into the databases. The characteristic tokens are 'model', 'form', 'field', 'class'.

**Java Map** topic—communication on java map structure and its aspects. The characteristic words are 'value', 'list', 'use', 'string'.

**Learning** topic—posts on coding skills development and IT education. The characteristic tokens are 'use', 'good', 'time', 'would'.

**Errors** topic's characteristic words are 'try', 'error', 'get', 'work'.

**Testing** topic's characteristic words are 'test', 'use', 'selenium', 'run'.

**Servers** topic's characteristic words are 'django', 'app', 'server', 'use'.

We did not expect to see Learning and Java Map topics in the list of the selected features of RF, since they seem to be less related to Selenium package dataset. In this sense LR model optimization led to a more expected feature subset.

**hSBM topics interpretation.** We interpret top 3 features of CatBoost and Random Forest models, hSBM feature space. We interpret topics by providing their characteristic tokens with associated probabilities in Table 6.

We notice that Topic 641 looks similar to LDA Testing topic with 2 characteristic tokens overlapping ('test' and 'run'). Additionally, Topic 641 contains a common 'server' token with LDA Servers.

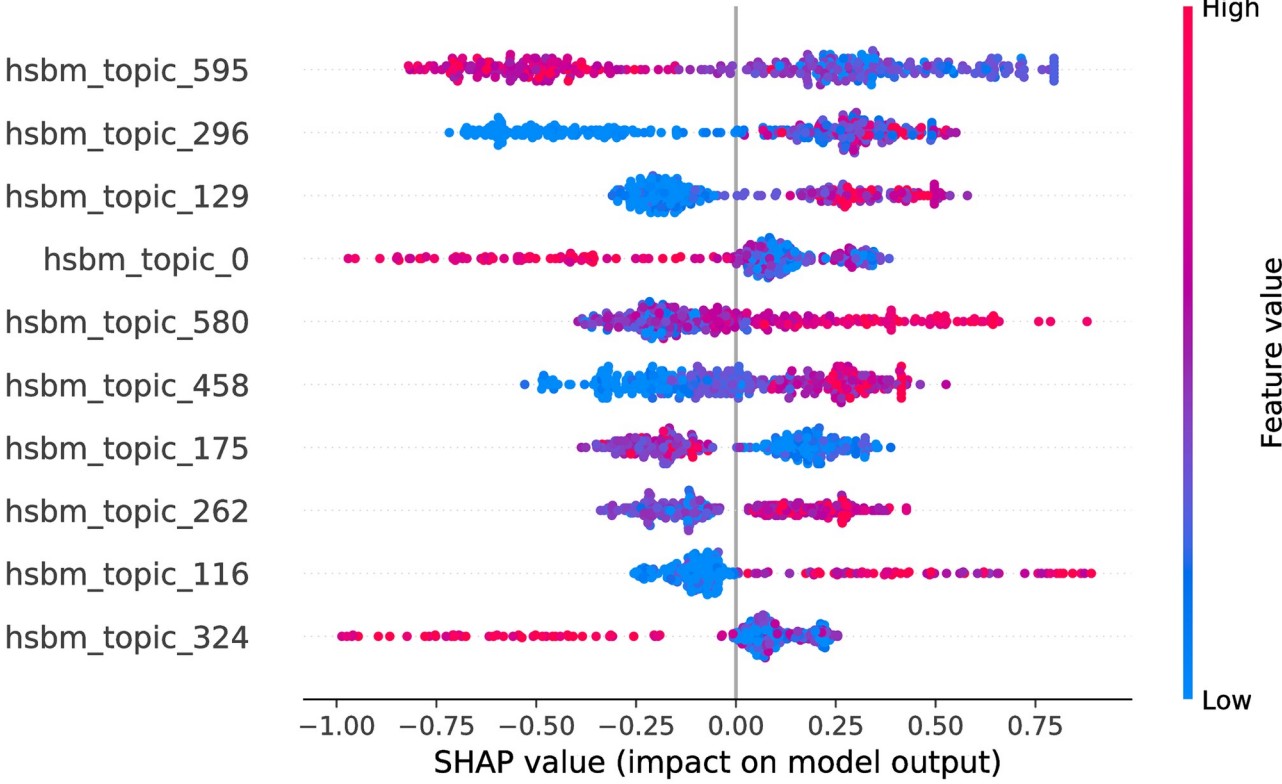

**Fig 12. CatBoost model SHAP values, hSBM feature space.** The figure illustrates the influence of the features on the output of the model fitted on Selenium package, minor updates, event-based time steps dataset. Color encodes the feature value and the X axis represents the impact of the feature in a particular case. Only top 10 features from the feature space are shown.

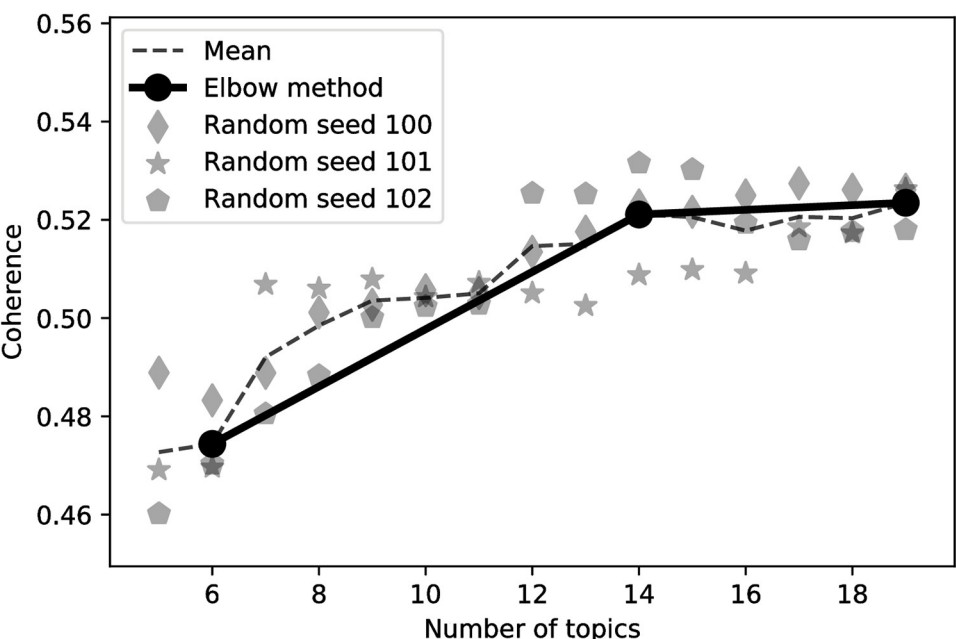

**Fig 13. Number of LDA topics optimization.** The figure demonstrates coherence measures for models randomly initialized with 3 random seeds, as well as mean coherence across the models. Elbow heuristic is applied using the mean coherence across the models. As an outcome, 14 LDA topics model is used throughout the study.

**Table 6. hSBM topic tokens.**

| Topic 595 | | Topic 296 | | Topic 129 | | Topic 519 | | Topic 641 | |
|---|---|---|---|---|---|---|---|---|---|
| **Token** | **Prob.** | **Token** | **Prob.** | **Token** | **Prob.** | **Token** | **Prob.** | **Token** | **Prob.** |
| attribute | 0.28 | setting | 0.98 | team | 0.28 | driver | 0.45 | run | 0.089 |
| item | 0.18 | settingsi | 0.0015 | game | 0.15 | protractor | 0.097 | test | 0.064 |
| insert | 0.10 | settingsand | 0.0011 | player | 0.11 | phantomjs | 0.088 | com | 0.051 |
| parent | 0.087 | yourproject | 0.0010 | story | 0.11 | headless | 0.076 | server | 0.047 |

The table provides characteristic tokens with associated probabilities for top 3 features of RF and CB models. Since topic 595 is common for both models, there are 5 features interpreted in total.

### Synthetic data results

**Response strength analysis.**  We assess performance of the three estimators on the synthetic data for a varying fraction of event-related posts. We perform permutation tests for all the instances and plot the results as a scatter plot (Fig 14). Error bars show 0.95 confidence intervals for the multiple instances of the same configuration. The color indicates the maximum p-value among the 15 instances of the same config—the darker, the smaller the p-value is. One can see that the first significant p-value is at 0.25 fraction of the event-related messages. Significance of 0.05 and 0.01 is reached by CB and RF, respectively.

In terms of absolute performance, all estimators perform comparably—CIs overlap, consequently there is no significant difference in their performance.

## Discussion

### Results interpretation

We found 13 significant fits of LR models based on the Log-likelihood Ratio test after applying the corrections for multiple comparisons. At the same time we observed 7 models with significant permutation test results in the SO data. The null hypothesis can be rejected—we have found a statistically significant change in the topics distribution when events take place. This can be already stated on the basis of the significance of model fit. Moreover, we have demonstrated that the fitted models generalize to the unseen data. Even though the tests indicate significant results, the absolute model performance is rather weak. We feel that there is a need for model improvement before it is used for solving real-world challenges.

### Limitations

**Time step design.**  The event-based time steps might violate the independence assumption for the logistic regression model in case there are multiple updates of the same type taking place within the time interval of a time step. It leads to using some fraction of posts in multiple entries, making the entries dependent. The violation does not allow to rely on the logistic regression model results in case of the multiple packages dataset. However, this setting can be used in the machine learning approach with its own benefits, such as more data points and a narrower-defined setting in comparison to the calendar week-based time steps.

The proposed time step representation has low risk of encountering ethical issues because it avoids having to label individual messages and hence it can better preserve privacy of individual message contributors.

**Response lag.**  In this work we assume that the reaction can be observed within up to 7 days after the event took place. This assumption was empirically derived from the

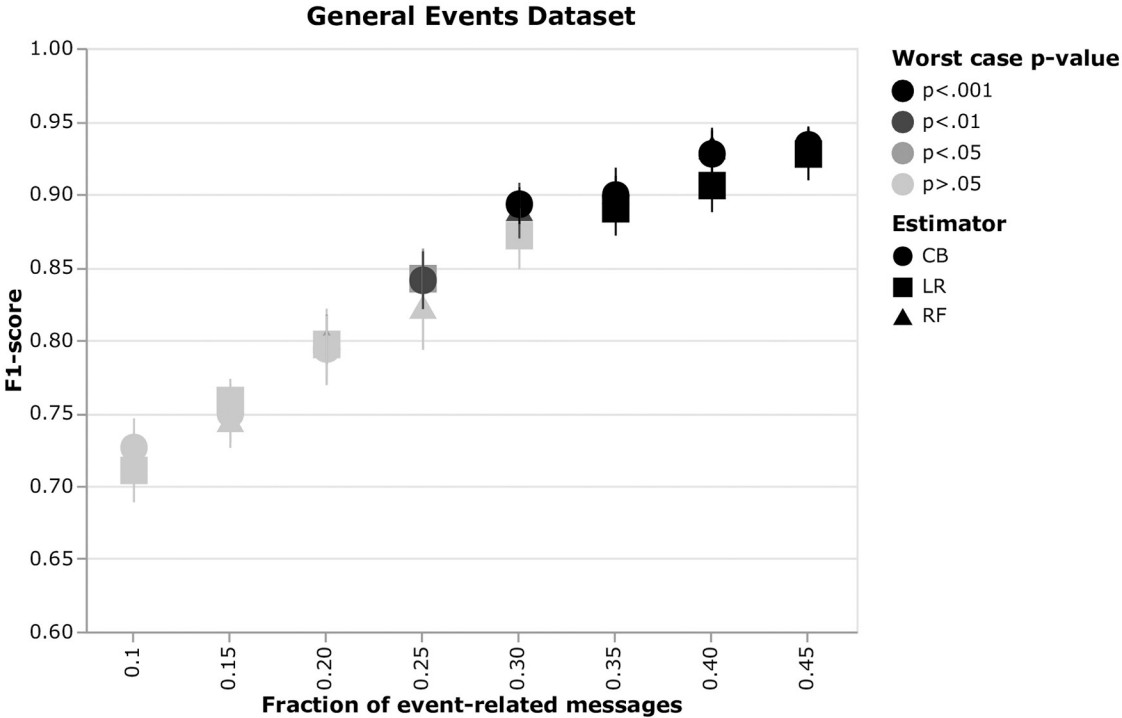

**Fig 14. Estimator performance on the synthetic data.** Performance of CatBoost (CB), Random Forest (RF) and Logistic Regression (LR) estimators versus a fraction of the event-related messages. The experiments were conducted on a synthetically generated dataset. Performance results are means over 15 randomly initialized dataset instances of the same configuration. The illustrated p-values are obtained from the permutation tests (1000 permutations) and are the maxima (as the worst case) over the 15 random initializations. The error bars account for 0.95 confidence intervals computed from the random initializations.

sparsity of the events-posts as well as our understanding of the software engineering workflows. Moreover, when designing reference time steps, one has to assume that the reaction to the most recent event is not in the data any longer. For the event-based time steps, we assume that day 8 after the event would belong to the reference time step. The assumption becomes unrealistically strict when considering daily time steps. Our exploratory experiments on the daily time steps confirm that. Depending on the available length of the time series, nature of events, and activity of the community, the time step window can be optimized.

It should be noted that the two suggested time step designs are aimed at two different temporal natures of the reactions. The event-based design is aimed at the posterior reactions to events, and the calendar week based is aimed at both prior and posterior. It should be taken into account if comparing the results.

**NLP tools.** We are aware of the limitations of the sentiment analysis tools [46, 47], as well as part-of-speech recognition tools [48] applied to the SE domain. Indeed, it makes SE texts challenging for the NLP tasks. We used a classical NLTK python package POS tagger and NLTK Vader sentiment analysis over other potentially more powerful methods like deep learning attention-based [49] and manual labeling-based [50] to ensure the best generality possible. While the deep learning approach is computationally intensive, especially when applied to large datasets. The manually labeled dataset is available for SE texts, however, there is no guarantee that there is a similar tool in other domains. Concluding: there is definitely space to improve the approach, which we discuss below.

When generating the event-related synthetic messages, we assume that the three community components are represented by nouns and the change operations—by verbs. Depending on the community or its language, it might not hold.

**Generator fine-tuning.**  To fine-tune the generator, we use the multiple packages post sample, which contains both event-related and not related entries. To generate event-related and background messages separately, we use the seed phrases, designed based on the 3-Compound model. This way we overcome the requirement of manual labeling of the messages, making the approach scalable and applicable to the content where possibilities for the manual labeling are limited.

**Design decisions.**  Multiple design decisions were made throughout the study. Ideally, all the applied thresholds should be further optimized to find an optimal configuration and evaluate their influence on the final result. For instance, the 1% threshold on inclusion of the package-related messages into the dataset might be sub-optimal, however, its optimization would require an additional correction for multiple comparisons. At the same time, this decision does not affect the single-package dataset configurations. We have done the optimization where it was computationally and statistically feasible.

The number of LDA topics in the discussed papers by Barua et al. [12], Yang et al. [13] and Abellatif et al. [14] is 40, 30 and 12, respectively. This is the same order as in our work.

## Implications for practitioners

In the current state the approach requires adjustment for applying for real world problems. Concretely: model performance at this point is unsatisfactory for reliable event detection and has to be improved.

There would be several benefits from the reliable detection of micro-events motivating the continuation of this work, as it would: (1) enable software developers to observe the event-related community interactions. (2) On historical data, the developers would be able to make better informed decisions when choosing dependencies for their projects (i.e. identify problematic dependencies). (3) Finally, it would boost the feedback loop between users and the developers—the event-related interactions can help measure a release's success.

More broadly, the approach might benefit other settings, like detection of emerging scams, illegal products, fraud schemes, etc. Generally, any setting with micro-events and linked forum-like textual communications might make use of the current study.

## Future work

As it was mentioned above, the pipeline can be improved, for example by adjusting the NLP tools to the domain. Using domain-agnostic tools, we have successfully demonstrated that micro-event detection is possible in the challenging domain of SE. Considering the improvements, there are more advanced extended LDA models, such as Author-Topic LDA, which links authors, topics, documents and words [51], LDA with Genetic Algorithm, acting on multimodal data [52] or LACT acting on the source code [53]. We feel that specifically for SE, source code analysis might contribute towards model performance improvement.

In the study we have investigated two pooling approaches: average across the messages within the time step and more advanced statistics with min, max and mean value for each LDA topic. We considered the advanced pooling approach as secondary analysis and observed no performance improvement on the test data. There might exist more perspective approaches, using, for instance, clustering prior to pooling to get more fine-grained statistics. Finally, it might be worth investigating more in detail other time window designs, as well as adding an additional lag between the event and reference time windows.

As an alternative data representation, deep learning unsupervised models can be used. For instance, variable input size autoencoders [54], and context-aware word embeddings from transformer-based deep learning architectures [43].

Finally, the synthetic dataset generation pipeline may be improved by looking into more advanced ways of comparing real world and synthetic data. Potential directions are: application of alternative distance measures in parallel with the used ones, and use of human intelligence for assessment.

## Conclusions

The contribution of the paper is a feasibility study of the micro-events detection on the example of the FLOSS version releases. We have demonstrated significant performance of the proposed approach for micro-event detection and rejected the null hypothesis.

We lay out a detailed analysis to understand model decisions. The experiments on the synthetic datasets help understand the limitations on the detectability of the studied micro-events. The analysis of the features contributing to our models can help understand better the nature of the predicted events, and contribute insight for the monitoring and management of the FLOSS ecosystems health. Finally, we identify a series of limitations in the message/time step representations, models and data preparation that lead to several potential lines of future work.

## Supporting information

**S1 Appendix. Goodness of fit measures of Logistic Regression models.**
(PDF)

**S2 Appendix. Detailed performance of estimators.**
(PDF)

**S3 Appendix. Synthetic data generator optimization.**
(PDF)

## Author Contributions

**Conceptualization:** Artur Sokolovsky, Thomas Gross.

**Data curation:** Artur Sokolovsky.

**Formal analysis:** Artur Sokolovsky.

**Investigation:** Artur Sokolovsky.

**Methodology:** Artur Sokolovsky, Thomas Gross, Jaume Bacardit.

**Software:** Artur Sokolovsky.

**Supervision:** Thomas Gross, Jaume Bacardit.

**Writing – original draft:** Artur Sokolovsky.

**Writing – review & editing:** Artur Sokolovsky, Jaume Bacardit.

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
