## [Decision Letter · Decision Letter 0]

2 Sep 2020

PONE-D-20-18285

Detection of FLOSS version release events from Stack Overflow message data

PLOS ONE

Dear Dr. Sokolovsky,

Thank you for submitting your manuscript to PLOS ONE. After careful consideration, we feel that it has merit but does not fully meet PLOS ONE’s publication criteria as it currently stands. Therefore, we invite you to submit a revised version of the manuscript that addresses the points raised during the review process.

We look forward to receiving your revised manuscript.

Kind regards,

Jacopo Soldani

Academic Editor

PLOS ONE

Journal Requirements:

2. Please clarify whether all terms and conditions of the websites, software and datasets used were complied with in the data collection and data sharing processes.

3.   We note that Figures in your submission contain copyrighted images. All PLOS content is published under the Creative Commons Attribution License (CC BY 4.0), which means that the manuscript, images, and Supporting Information files will be freely available online, and any third party is permitted to access, download, copy, distribute, and use these materials in any way, even commercially, with proper attribution. For more information, see our copyright guidelines: http://journals.plos.org/plosone/s/licenses-and-copyright.

a)   You may seek permission from the original copyright holder of Figure(s) [#] to publish the content specifically under the CC BY 4.0 license.

b)   If you are unable to obtain permission from the original copyright holder to publish these figures under the CC BY 4.0 license or if the copyright holder’s requirements are incompatible with the CC BY 4.0 license, please either i) remove the figure or ii) supply a replacement figure that complies with the CC BY 4.0 license. Please check copyright information on all replacement figures and update the figure caption with source information. If applicable, please specify in the figure caption text when a figure is similar but not identical to the original image and is therefore for illustrative purposes only

Reviewers' comments:

Reviewer's Responses to Questions

**Comments to the Author**

1. Is the manuscript technically sound, and do the data support the conclusions?

Reviewer #1: Yes

Reviewer #2: Partly

2. Has the statistical analysis been performed appropriately and rigorously? 

Reviewer #1: Yes

Reviewer #2: Yes

3. Have the authors made all data underlying the findings in their manuscript fully available?

Reviewer #1: Yes

Reviewer #2: No

4. Is the manuscript presented in an intelligible fashion and written in standard English?

Reviewer #1: No

Reviewer #2: No

5. Review Comments to the Author

Reviewer #1: This article proposes a logistic regression based models for the detection of micro-events from textual data using messages from SO Q&A platform to detect FLOSS version release events. The LR models utilise a feature space composed of a selected number of LDA topics and sentiment analysis features. The article is fairly well-written, particularly the Introduction and Background material which provide an ample literature review of related work. The article also provides sufficient descriptions of the models’ statistics, evaluation and analysis, which seemed to have been performed to a good technical standard. Overall, it is obvious to me that the reported study involves a significant and technically sound work, and deserves publication. However, the manuscript also needs a fair amount of revision (which in my opinion is more than a minor revision) to address the following observations:

• The article requires improvement in terms of structure, organisation and flow of information. As it is, the manuscript falls short of providing a clear and easy to follow structure of the study and proposed models/processes, and the reader has to read the article multiple times to obtain a full-picture of the whole reported work and how each process leads to the next. The descriptions seem to go back and forth in places and, hence, the authors need to make sure that the article provides, with the help of better illustrations, simple description of the various phases of the study, their sequences and associated processes, and how each leads to the next.

• The article in my opinion needs some focus in its aim: initially, and as the title indicates, the focus seems to be on developing models for a successful micro-events detector/estimator. However, later on and once the performance of the estimators was reported to be insignificant, the focus seems to shift predominantly to ‘better understand the data’ and generating synthetic data to find the detectability threshold. Again, I would like to emphasis the quality and standard of presented work here, however the authors should also declare a clear aim/focus of this article one way or another right from the beginning, reflect that in tile properly and stick to this aim.

• The developed synthetic data generator is one of the main contributions of this work. However, I am not sure how such a finely tuned and controlled synthetic dataset is going to help the development of a detector that generalises to a non-synthetic domain without fine-tuning on any real world data.

• Relating to above, I would have liked to see the authors focusing more on improving the performance of the proposed detector pipeline via, for example, attempting and comparing other topic modelling techniques such as LSA, pLSA and Deep LDA, etc., and adjusting the NLP tools of the sentiment analysis to targeted domain (issues the authors have already discussed as limitations of the study). In this respect, the authors should also try to deviate a bit from their target to limit the number of variables in the feature space (which seems to me solely based on the claim of ‘curse of dimensionality’, which is known to be dataset and algorithm dependent and has well-established remedies).

• Manuscript contains few typos and I suggest the authors proofread it again and correct (quick example is on p.8 in the Model Performance Section: ‘… we compute a mean of PR-AUC - event as a positive label and no-event is a positive label.’)

• I have noted that this exact and full manuscript currently appears on: 1) the arXiv archive by Cornell University ( at https://arxiv.org/abs/2003.14257), and on researchgate.net (at https://www.researchgate.net/publication/340331989_Detection_of_FLOSS_version_release_events_from_Stack_Overflow_message_data), and would like to advise the authors to check with PLOS ONE publications policy and terms and conditions.

Reviewer #2: The paper provides a methodology to detect floss version releases through an event detection process. Overall, the process of data collection, pre-processing steps, feature construction (i.e. sentiments, topics) and the ML algorithms used to evaluate the event detection process are presented clearly. Additionally, the event detection time window formation is understandable. However, some points need to be improved:

1. The introduction is rather like a report and contains different parts of information. I suggest the authors to improve the flow in the text and connect with logical cohesion the different parts.

2. Also, it is not mandatory, but most research approaches form more than one RQs. You could reform the structure and add more research questions i.e. evaluating which is the best ML algorithm to perform event detection in SO post.

3. Similar to the introduction section also the “Background material and related work” section needs better connection between paragraphs.

4. Additionally, for the related work it could be helpful for readers to add a comparison between different algorithms used to identify events from different online sources such as the Twitter (i.e. https://dl.acm.org/doi/10.1145/1978942.1978975). Identify which is the target source, which is the data analysis method, which option of event detection authors used etc. and compare with your own research.

5. Additionally for the bibliography part there are many works which use LDA in Stack Overflow posts (e.g., https://link.springer.com/article/10.1007/s10664-012-9231-y or https://link.springer.com/article/10.1007/s11390-016-1672-0). I would expect to see some discussion for this in the bibliography part.

6. Moreover there are work which identify from Stack Overflow posts technology releases i.e. http://das.encs.concordia.ca/uploads/abdellatif_msr2020.pdf (fig 3). So, I would expect to see a refence for this as well.

7. In “Pre-processing” subsection for sentiment analysis calculation please explain why you select to use Vader method and not use a sentiment analysis approach which is constructed for developers’ post such as https://link.springer.com/article/10.1007/s10664-017-9546-9

8. Additionally, in the same subsection you can add a graph which depicts the results of coherence measure used to select the appropriate number of topics (i.e. https://link.springer.com/article/10.1007/s10664-020-09819-6 see Fig. 12)

9. In the “Analysis pipelines” subsection the analysis for the construction of Logistic Regression and Random Forest is good. However why do you select this ML method? How about an SVM approach?

10. I suggest adding a section after “Limitation” section with implications for practitioners. For example, use case scenarios on how someone can use your research approach to find a new FLOSS version.

6. PLOS authors have the option to publish the peer review history of their article (what does this mean?). If published, this will include your full peer review and any attached files.

Reviewer #1: No

Reviewer #2: No

---

## [Author Response · Author response to Decision Letter 0]

13 Nov 2020

We are responding to the reviewers and editor comments in a separate document which was uploaded earlier.

---

## [Decision Letter · Decision Letter 1]

20 Jan 2021

Is it feasible to detect FLOSS version release events from textual messages? A case study on Stack Overflow

PONE-D-20-18285R1

Dear Dr. Sokolovsky,

We’re pleased to inform you that your manuscript has been judged scientifically suitable for publication and will be formally accepted for publication once it meets all outstanding technical requirements.

Kind regards,

Jacopo Soldani

Academic Editor

PLOS ONE

- - - - - - - - - - - - - - - - - 

Reviewers' comments:

Reviewer's Responses to Questions

**Comments to the Author**

1. If the authors have adequately addressed your comments raised in a previous round of review and you feel that this manuscript is now acceptable for publication, you may indicate that here to bypass the “Comments to the Author” section, enter your conflict of interest statement in the “Confidential to Editor” section, and submit your "Accept" recommendation.

Reviewer #1: All comments have been addressed

Reviewer #2: All comments have been addressed

2. Is the manuscript technically sound, and do the data support the conclusions?

Reviewer #1: Yes

Reviewer #2: Yes

3. Has the statistical analysis been performed appropriately and rigorously? 

Reviewer #1: Yes

Reviewer #2: Yes

4. Have the authors made all data underlying the findings in their manuscript fully available?

Reviewer #1: Yes

Reviewer #2: No

5. Is the manuscript presented in an intelligible fashion and written in standard English?

Reviewer #1: Yes

Reviewer #2: Yes

6. Review Comments to the Author

Reviewer #1: Thank you for addressing my comments and, yes, your paper has now been improved technically and readability wise.

Reviewer #2: I checked that authors fullfilled all my comments, even the one with the topics evolution. So I believe that their work is acceptable now.

7. PLOS authors have the option to publish the peer review history of their article (what does this mean?). If published, this will include your full peer review and any attached files.

Reviewer #1: **Yes: **Professor Abdulhussain E. Mahdi (known as Hussain Mahdi)

Reviewer #2: No

---

## [Editor Report · Acceptance letter]

22 Jan 2021

PONE-D-20-18285R1 

Is it feasible to detect FLOSS version release events from textual messages? A case study on Stack Overflow  

Dear Dr. Sokolovsky:

I'm pleased to inform you that your manuscript has been deemed suitable for publication in PLOS ONE. Congratulations! Your manuscript is now with our production department. 

Kind regards, 

on behalf of

Dr. Jacopo Soldani 

Academic Editor

PLOS ONE